## Book blurb:

*A Pragmatic Account of Cognition: Rethinking Externalism and Intentionality*
        Under contract with Bloomsbury Press to be delivered August 2023

The book advances a theory of cognition that challenges a central assumption of the mainstream view that human cognitive prowess rests in a biologically fundamental capacity for representation. I argue that this reductive internalist picture is mistaken, misguided, and ultimately misguid*ing* and in its stead, I advocate for a version of externalism on which representation *use* is a learned skill. In rejecting the standard representational view my approach has many sympathies with embodied, enactive, extended cognitive accounts but it diverges from them in important ways as well, most saliently in agreeing that representation use is indeed a central aspect of high-level, cognitive behaviour. Thus, my approach is an attempt to make sense of and unify the important insights of the various traditions while eliminating the commitments that hold us back. The second half of the book provides a comprehensive, inter-disciplinary survey of the empirical support and challenges for externalism focusing in particular on recent work in computer science, comparative psychology, and cognitive psychology.

**Excerpt pp. 137-145**: Salay. (2019). Learning how to represent: An associationist account. *The Journal of Mind and Behavior, Vol. 40*(2), pp. 121-145.

### The Language Tool

As in the smoke/fire example I've been using, a sign vehicle might be some phenomenon that naturally co-occurs with an unconditioned stimulus or it might be some phenomenon that naturally co-occurs with an unconditioned response. Some animal alarm calls seem to work like this. Flagging, when an animal lifts its tail and displays the white fur beneath, serves to trigger the flight response in nearby animals, presumably because the white fur is also exposed during the flight response itself, as the troupe of animals is fleeing. Of course, flagging and many other animal alarm calls are often innately driven significatory actions and, as such, lie at the inflexible end of the representation spectrum.

The more sensorily distant a sign vehicle is from a stimulus or a response, the more explicitly it must be learned. Smoke is a feature that naturally occurs when fires are present and the white fur underneath a raised tail is a natural feature of a fleeing situation, but the more flexible an organism is, the more arbitrary sign vehicles it can learn. Bonobos, for example, have a wide repertoire of gestures and vocalisations, some of which seem to be grounded in the sort of natural relation described above, such as alarm calling, and others that seem to be locally learned, ontogenetic ritualisations that serve as sign vehicles. Infant-mother dyads, for example, will develop idiosyncratic, stylised carry signals — shoulder touches — that trigger responses — carrying behaviour — that would normally be triggered by overt carry-request behaviour — climbing onto mother's back. (Halina, Rossano, and Tomasello, 2013).

Initially perhaps, if we are developing an account of human language development, gestural and vocal sign vehicles are closely linked, sensorily, to their objects; they are naturally co-occurring features of stimulus or response situations. But, as in the case of bonobos, increasingly abstract short-hands for these natural sign vehicles develop. The use of onomatopoeic words and stylised gestures, for example, takes us one step further along the representation continuum; increasingly arbitrary sign vehicles follow. A human child must learn, through a great deal of (often explicit) teaching/learning experience, to respond to the utterance "'kæt" as she would to the presence of an actual cat. Mother, pointing to the cat says, "Cat. Look at the *Cat.* Do you see the *Cat*? What colour is the *Cat*?" and so on. We now have good evidence that the more of this teaching/learning experience a child has with words, the more sophisticated her language comprehension and use will be; and, the stronger many other cognitive skills will be as well. (Haak, Downer, and Reeve, 2012; Sénéchal and LeFevre, 2002; Suskind, 2015; Swanson, Orosco, and Lussier, 2015; Yeong, Fletcher, and Bayliss, 2017)

As has been observed (Chomsky, 1968; Davidson, 1975; Hockett, 1977), language can be rigorously distinguished from animal signaling of the sort I've been describing thus far by, at least, two key features: compositionality and productivity. Animal communication systems seem to be made up of units that can neither decompose nor be combined, or, if they can, only to limited degrees and in fixed ways. Likewise they do not seem to be productive, with the repertoire of calls for a given animal being more or less fixed for its lifetime. These observations support the internalist hypothesis that some innate factor, beyond flexibility and fluidity, must account for language development.

Hauser, Chomsky, and Fitch (2002) have suggested that an innate recursive capacity is required to explain our productive use of language. If a capacity for recursion entails a capacity for decomposition, for seeing the parts that make up a whole, some studies (Van Leeuwen, Verstijnen, and Hekkeit, 1999) seem to indicate that humans are lacking in this quarter: while we are adept at mentally combining images — imagine a horse with a horn — we are poor at mentally decomposing them — rotate the bottom right quadrant of an image you are remembering. The finding is that decomposition of this sort requires continual, sensorimotor interaction with the object under analysis. There is some evidence, then, that a capacity for recursion might not be an innate biological feature of the modern human. Perhaps the simpler capacity for combining alone could account for the compositionality and productivity of the language tool we have developed. We see a strong inclination to exercise this capacity in modern human children who seem compelled to stack blocks, pile sticks, and group objects — as well as in other animals: if there are objects to be moved and manipulated, an orangutan will unfailingly bring them together; birds and chimps stack twigs; and, octopi combine objects to solve problems.

But whichever internal factor it is that plays this crucial role in the development of language, we run the risk of becoming side-lined by reductive internalism, again, if we focus solely on a capacity-based explanation. A more comprehensive account will look at the subtle and gradual co-development of sign vehicle use, on the one hand, and, of the sign vehicles themselves, on the other. Consider: the more specific a symbol is, that is, the more it is tied to a particular response, the more constraints there will be for usage and, consequently, the less use it will be in combination with other sign vehicles. I can use smoke, as a sign vehicle, in a number of ways, but those ways are quite constrained toward signaling emergencies. The ease with which a sign vehicle can be produced must also play a role here. Producing smoke is a rather laborious process. If one wanted to combine this sign with another, also difficult to produce, combining the two would augment those difficulties, making combining less practicable and therefore less likely to occur. The fewer usage constraints there are and the easier sign vehicles are to produce, on the other hand, the more a user will be drawn to combine them. Words, as sign vehicles, are ideal in this regard (for humans).

Some excellent work has been done to begin laying down the theoretical framework within which we can better investigate the complex, dynamic processes that govern language use and language development. Deacon (2011), for example, has introduced some new concepts to help us theorise about how entirely new kinds of processes emerge out of complex, cyclic, dynamics of this sort. From the other side of the fence, as it were, Christiansen and Chater (2008) suggest that language itself should be seen as an "organism" that evolves in response to selectional pressures. And, to name just one more from a growing number, Jeffares (2010) argues that our capacity to develop cognitive-enhancing tools rests on the scaffolding provided by earlier cycles of more primitive tool use. The central point here is that we should be looking both internally *and* externally in order to explain the features of language and language users.

It is instructive here to look at the way in which other animals use sign vehicles. Consider a typical interaction with Kanzi,[1] a 37-year-old bonobo who has learned to respond to hundreds of words and can use lexigrams[2] flexibly in order to communicate with humans. Kanzi, being a full grown male bonobo, often has energy to burn. After a morning of word play in the research centre, he is restless and twitchy. There are many games that he likes to play, but one of his favourites is *ball*. Imagine Kanzi as he paces about the room. There is a heightened expectation for *ball* as a result of his restlessness — he is a fluid system — and so the lexigram for *ball* is a ready-to-hand tool for him in this moment. He picks it out from the sheets of lexigrams that lie

---

[1] I draw, here, on Savage-Rumbaugh's (1990) descriptions of Kanzi.

[2] Lexigrams are visual symbols for words that are used in language instruction with non-verbal animals such as great apes.

about the room, points at it, and looks around expectantly. When someone responds, perhaps by looking in his direction, perhaps by saying, "What would you like to do Kanzi?" new possibilities open up. Kanzi points again to the lexigram and then to his interlocutor, indicating that they should go and play ball. As with *A*, there is no need to suppose that Kanzi's lexigram use is an outward sign of some inner representational life, that Kanzi already has "in mind" the desire to play and merely uses the lexigrams to communicate this desire. On the externalist view being developed here, there are no inner representations to be communicated at all. Kanzi is drawn, according to myriad attentional pushes and pulls that are continually shaping, dissolving, and reshaping, to whatever is ready-to hand in his environment, itself a product of Kanzi's past learning experiences and what is currently winning out in the competition for attention. Here, it is restlessness, excessive energy, that dominate, and so Kanzi is drawn to the lexigram for *ball*. There is no plan here, no thought, "Now I have to find someone to play ball *with*;" rather, because balls are always thrown *to* someone or caught *from* someone, when Kanzi sees a person, he sees a potential playmate.

Now consider that these sign vehicles are tools, cognitive-enhancing tools to be sure, but, more basically, they are tools for effecting change in one's environment: by yelling "fire" I can induce panic in a group of people; by making a request, I can acquire food. Seen in this light, they are new additions to what is ready-to-hand. As one gains facility with these tools, as they become increasingly ready-to-hand, the more they open up the space of possibilities. With them one can plan for the future — as Kanzi does when he sets up the conditions for ball playing — and refer back to the past — as Kanzi does when he points to a bite on his arm and signs "Matata bite."

But if we have no internal, innate, capacity for representation at all, we might wonder, how could this sign vehicle skill ever develop into *conscious* R-activity, into thoughts and conscious perceptions? Giving a detailed answer to this question takes us well beyond the scope of this paper, but in the interest of completeness, I will describe, in broad strokes, how I see this part of the account unfolding.

As soon as Kanzi "says" "you ball me," the utterance is now a thing in the world. It is some situation to which we can react. One could nod in assent, one could begin kicking the ball, or, one could describe the situation: "*You* say, 'you ball me.'" Describing things is, in itself, a thing to *do*. This meta-activity can continue: "*You* say, 'you say, "you ball me".'" And so on. Of course, except in certain contexts (philosophical ones, perhaps), there isn't much motivation for continuing in this way: the pushes and pulls of our past experiences as they unfold in the present environment will generally out-motivate such meta-meta-activity. On the other hand, our language-rich environment, overflowing with sign vehicles, presents an enormous pull to engage in this descriptive, meta-activity. Indeed, though they are not visible — sign vehicles are not something we *see* (unless they are words on a page, of course) — the environment of a literate, adult human contains words more than it contains anything else. This is quite remarkable, particularly so when we consider how unaware we are of this singular fact. If the phenomenological picture of ourselves as beings-in-the-world is correct, then, when we augment it with a world that is replete with sign vehicles, the traditional Cartesian mind-body dualism comes into soft relief: we are beings who, most fundamentally, act; however, as it happens, most of our tools for action are tools of reason, namely, words.

In bare perception, there is just action, a perceiving of the world. There is no self-awareness, no consciousness at all. As we become skilled language users and as our landscape becomes filled with sign vehicles, increasingly we use them to describe, to ourselves, the situations we are in. These self-descriptions, what we typically call "conscious thoughts," are not new things-in-the-world: they are simply descriptions, and, consequently, not items to be added to our ontology. But the mistake is easily made. Consider Fred, admiring a red peony in his garden. In bare perception, he simply perceives the flower. He is not aware of the "content" of this perception, of what the perception is about, as something separate from this flower in front of him. But now Sally sidles alongside and says, "I wonder what that red is like for you. For me it is so firey!" Fred frowns, perplexed. Sally, we might say, has just used some words to describe *her perception of the peony*. When we put the matter like this, it seems as though we now have something extra here, Sally's experience of her perception, what we sometimes call the qualia of her perception. But we don't have anything extra here at all. The thought that we do is a consequence of loose or sloppy describing. Sally has used some words to describe her description of the peony, not her perception of the peony. In bare perception, Sally is simply perceiving the flower as Fred is; she has no awareness of this at all. In conscious perception, however, Sally is

using words to describe the situation — the peony perceiving — to herself. This is a description. We often call this description the perception itself, but on the externalist account being developed here, we distinguish between perceptions and descriptions of perceptions. When Sally uses words to report on this description to Fred, she is describing the *description*, not the perception.

Because descriptions lend themselves to reification (we can describe our descriptions), they lead to this sort of dualistic mistake, of supposing that there is some experience in addition to the bare perception. Descriptions mislead us in a second way as well. Since we must use the sign vehicles we have on hand, our descriptions will be shaped and constrained by the words we happen to have. The concepts we develop out of these descriptions, *self*, *personality, mind*, for example, are as much a function of the words we have as they are of the world itself.

On this view, then, it is the compounding effect of sign vehicle use and the proliferation of the vehicles themselves from which conscious thought — self-descriptions — emerges. This position is clearly sympathetic with those who see qualia talk as fundamentally misguided (Churchland, 1985; Dennett, 2001), but, unlike those, it offers an explanation for why thoughts, feelings, and emotions seem to have a separate existence. Thus, while this externalist view is firmly grounded in materialism, it also offers an explanation of the psychological draw toward dualism, something most reductive internalist views fail to address.

To summarise, then, when a flexible/fluid system develops a capacity for using sign vehicles, first in the very rudimentary way in which animals use distress calls to warn of danger, and then, in more sophisticated ways, using increasingly general and arbitrary sign vehicles to perform increasingly abstract actions, the beginning of R-activity emerges. The dynamic interchange between representational tools that are easily produced and combined with some internal capacity/inclination to combine them underwrites an explosive proliferation of vehicle and vehicle use. Over time, the world into which new humans are born becomes a language-rich one. A child's development into a literate adult is thus a symbiosis between its own flexibility and fluidity and the word-filled landscape in which it grows. How this skill with words eventually develops into thought and self-consciousness, what understanding amounts to, what distinguishes propositional knowledge from skills are questions still to be addressed, though I have given an idea of the direction in which the explanation will lead.

## Conclusion

As the beginning of an externalist account of our representational capacities, this sketch might seem to raise more questions than it answers. Looming large for me, someone who has a particular interest in linguistics and the philosophy of language, is the uneasy awareness that in treating language solely as a "tool for representing," as this account seems to do, I have swept under the carpet the vastly complex ways in which language is in fact used, something Wittgenstein (1967) so evocatively and eloquently demonstrated in his *Philosophical Investigations*. And the big questions of consciousness remain: Is there something more to understanding the connection between CS and US/UR besides knowing how to exploit it? How do we explain the phenomenal aspects of thoughts and feelings? More generally, how does the language tool lead to self-consciousness? Here I have been concerned only to give a plausible externalist grounding of our representational capacities, not a full account of them. We must begin somewhere.

To sum up: R-activity is the cognitive capacity that cognitive science has been most actively investigating for the past 70 years. No surprise then that the computational/representational metaphor has been such a motivating force in the field, anti-representational challenges notwithstanding. But those challenges are ignored at our peril, since no account is forthcoming unless they are taken seriously. The externalist approach I have described here does that by grounding the representational part of R-activity in organism-level engagement with its environment, not in its neural activity. Thus, Dreyfus is correct to insist that, "being-in-the-world is more basic than thinking and solving problems; that it is not representational at all" (2007, p. 1146) but wrong to ignore the important consequences our representational tools have on the ways in which we can be-in-our-world. Marshall McLuhan's (1964) insight, that the medium is the message, promises to take on even deeper meaning as we explore this new role for language.

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
