# OpenReview forum: "The Language Tool"
_ICLR.cc/2022/Workshop/EmeCom — EmeCom Workshop at ICLR 2022_

### Official Review · Reviewer_qx34 · 2022-03-17
**The paper is not anonymous and the format could be improved. The paper is relevant to the workshop as it questions certain philosophical views (which inform AI research) about how humans engage with language.**

**Rating:** Weak rejection
**Confidence:** 3

**Review:**

#### 1 being lowest and 5 being highest
- Quality: 2 the form is a little sloppy (see Form below)
- Clarity: 3 the main takeaways are clear, but not everything is
- Potential to create discussions: 4 (see Relevance below)
- Originality: 4 within ML/AI the originality is high
- Significance of this work: - reviewer not in a position to judge


### Form:
This submission is not anonymous. The author’s name is visible at the very top of the pdf and as a footnote on every page.

The format is unusual and/or sloppy. For example:
- One would expect the submission to start with an abstract and use more section headings. Instead the submission starts with a ‘Book Blurb:’.
- The book blurb talks about ‘The second half of the book’, as if this were the foreword to a book rather than a paper.
- ’As in the smoke/fire example I've been using’. This is the first mention of the example. The paper could be improved by being more self-contained.

Because of these form issues I think the paper should be rejected.


### Summary of paper:
This philosophical paper discusses human and animal representational capacity through the use of sign vehicles (for example words or physical signs). The paper argues against purely internalist takes on representational capacity, which say that humans use sign vehicles to internally represent the world and be able to consciously think about the world. The paper gestures at a plausible externalist alternative explanation in which using words is mostly a way of interacting with the world, i.e. using words is just part of our action space (rather than at the core of our cognition).

The paper also briefly discusses some specific phenomena. For example, if animals have access to cheaper sign vehicles, then they are more likely to use and combine them. The paper further questions whether innate recursive capacity is needed for human use of language (as has been argued by others) as a capacity for recursion may entail a capacity for decomposition, and humans are bad at decomposition. The author suggests that the capacity for combining could explain many properties of human language (such as compositionality).


### Relevance (and potential to spark discussions):
At times computer scientists aim to develop agents that have similar language acquisition (and usage) properties to humans. The questions posed in this paper are relevant to the workshop as they question certain human language properties that have long been taken for granted.

---

### Official Review · Reviewer_qaLo · 2022-03-22
**Insightful essey on an alternative against the common internalist view of cognition, but fundamentally outside of the scope of this workshop**

**Rating:** Rejection
**Confidence:** 3

**Review:**

Interesting on language emergence under the lens of cognitive theory. It introduces the "externalist" view of language, which regards language and the abstractions that it tries to represent as a tool to be learnt, contrasting against the mainstream "internalist" approach which states that cognitive prowess rests in a biologically fundamental capacity for representation. This submission attacks the internalist take in favour of the proposed externalist view under different angles. This submission covers a broad range of topics in a "scattergun" approach by tackling multiple angles at once, yielding a confusing attack.

On its own merit, this submission can potentially add value to the field of cognitive theory, specially if its main points are further expressed in future writing. However, the nature of the conference for this submission is on artificial intelligence and representation learning therein. This paper fails to address the possible connection between the presented externalist view of cognition and how it could potentially be used to improve any field of AI related to emergent communication, leading to my proposed rejection based on the submission not being within the scope of this workshop / conference.

More specific comments:

The notion of R-activity is neither introduced nor cited, and for which I could not find a definition online, having either a definition or a citation can help readers like me.

The section "The language tool" (the second one) begins with "as in the smoke / fire example I've been using". Based on the submission alone, this example has not been introduced. I would encourage the author to re-state it, as otherwise readers are left wondering what that example actually is. The author does mention that this publication is an excerpt from other content, but further work could be put into improving this text as a standalone document. Perhaps not related. The last sentence of line 3 page 3 starts with "As with _A_...", with the concept of "A" in italics not having been introduced in the submitted text.

---

### Decision · Program_Chairs · 2022-03-25

**Decision:**

Accept

**Comment:**

We are deciding to accept this paper on the merits of it presenting a view of cognition and language that may be relevant to attendees of the emergent communication workshop.

We believe there may be some confusion between reviewers and author and the program chairs would like to apologize for causing this confusion and not being clear in our reviewing instructions about our novel guidelines for published work outside of ML. For this year's workshop, we were keen to have a variety of perspectives on emergent communication and invited submissions from many related fields such as cognitive science, philosophy, and biology. Our goal was to accommodate other fields with different publishing styles without giving undue burden to our reviewers to read very long papers or even book chapters. To entice submissions and get different perspectives we allowed authors to submit reasonable excerpts without any style changes. As specified in the abstract above, the submission is a blurb from a book and an excerpt from a paper. We should have specified that reviewers' main goal was to evaluate whether this would be a good starting point for interesting discussions with attendees and the guidelines for reviewing this work were different from regular unpublished papers.

After evaluating the reviews, it seems both reviewers agree that the ideas presented would be interesting for our community to discuss. The main reservations are about the form of the work but since our workshop is non-archival, this submission would mainly be for people interested in getting a background on the ideas before starting the discussion. That said, we can see how the ideas could be difficult to understand for readers in ML given the presentation. We can also make space for the author to post a full version of their paper if they believe that would be helpful and more accessible to attendees. When preparing for their discussion, we encourage the author to take into account an audience perspective that may be less familiar with their field and more grounded in ML. Addressing one reviewer's point about relevancy to ML, perhaps the discussions can be a cooperative effort between attendees and presenter to see where these ideas can be applied.